# Prediction of future visceral adiposity and application to cancer research: The Multiethnic Cohort Study

**Lynne R. Wilkens[1]\***, Ann M. Castelfranco[2], Kristine R. Monroe[3], Bruce S. Kristal[4], Iona Cheng[5], Gertraud Maskarinec[1], Meredith A. Hullar[6], Johanna W. Lampe[6], John A. Shepherd[1], Adrian A. Franke[1], Thomas Ernst[7], Loïc Le Marchand[1], Unhee Lim[1]

1 University of Hawaii Cancer Center, University of Hawaii at Manoa, Honolulu, Hawaii, United States of America, 2 Pacific Biosciences Research Center, University of Hawaii at Manoa, Honolulu, United States of America, 3 Keck School of Medicine, University of Southern California, Los Angeles, California, United States of America, 4 Jean Mayer USDA Human Nutrition Research Center on Aging at Tufts University, Boston, Massachusetts, United States of America, 5 School of Medicine, University of California San Francisco, San Francisco, California, United States of America, 6 Fred Hutchinson Cancer Center, Seattle, Washington, United States of America, 7 University of Maryland School of Medicine, Baltimore, Maryland, United States of America

\* Lynne@cc.hawaii.edu

**Data Availability Statement:** The data used for the current study are not publicly available due to privacy concerns and based on the consent from the study participants. However, data are available

## Abstract

### Background

We previously developed a prediction score for MRI-quantified abdominal visceral adipose tissue (VAT) based on concurrent measurements of height, body mass index (BMI), and nine blood biomarkers, for optimal performance in five racial/ethnic groups. Here we evaluated the VAT score for prediction of future VAT and examined if enhancement with additional biomarkers, lifestyle behavior information, and medical history improves the prediction.

### Methods

We examined 500 participants from the Multiethnic Cohort (MEC) with detailed data (age 50–66) collected 10 years prior to their MRI assessment of VAT. We generated three forecasted VAT prediction models: first by applying the original VAT equation to the past data on the predictors ("original"), second by refitting the past data on anthropometry and biomarkers ("refit"), and third by building a new prediction model based on the past data enhanced with lifestyle and medical history ("enhanced"). We compared the forecasted prediction scores to future VAT using the coefficient of determination ($R^2$). In independent nested case-control data in MEC, we applied the concurrent and forecasted VAT models to assess association of the scores with subsequent incident breast cancer (950 pairs) and colorectal cancer (831 pairs).

from the Multiethnic Cohort study (https://www.
uhcancercenter.org/for-researchers/mec-data-
sharing) to researchers who meet the criteria.

**Funding:** This research was funded by grants from
the US National Institutes of Health: P01 CA168530
(LRW, KRM, BSK, IC, GM, MAH, JWL, JAS, AAF,
TE, LLM, UL), U01 CA164973 (LRW, KRM, IC,
LLM), P30 CA071789 (LRW, GM, JAS, AAF, LLM,
UL), and P30 CA015704 (MAH, JWL) and USDA
the Agricultural Research Service under
Cooperative Agreement 58-8050-9-004 (BSK). The
funders (the National Institutes of Health and
USDA) had no role in study design, data collection
and analysis, decision to publish, or preparation of
the manuscript. Any opinions, findings,
conclusions, or recommendations expressed in
this publication are those of the authors and do not
necessarily reflect the views of the NIH or USDA.

**Competing interests:** We previously reported that
Dr. Kristal is a consultant for Metabolon, Dr. Ernst
is a consultant for KinetiCor, Inc., and neither of
these interests played any role in this study. We
confirm that Dr. Kristal's and Dr. Ernst's reported
interests do not alter our adherence to PLOS ONE
policies on sharing data and materials. The other
authors declare no potential conflicts of interest.

## Results

Compared to the VAT prediction by the concurrent VAT score ($R^2$ = 0.70 in men, 0.68 in women), the forecasted original VAT score ($R^2$ = 0.54, 0.48) performed better than past anthropometry alone ($R^2$ = 0.47, 0.40) or two published scores (VAI, METS-VF). The forecasted refit ($R^2$ = 0.61, 0.51) and enhanced ($R^2$ = 0.62, 0.55) VAT scores each showed slight improvements. Similar to the concurrent VAT score, the forecasted VAT scores were associated with breast cancer, but not colorectal cancer. Both the refit score (adjusted OR for tertile 3 vs. 1 = 1.27; 95% CI: 1.00–1.62) and enhanced score (1.27; 0.99–1.62) were associated with breast cancer independently of BMI.

## Conclusions

Predicted VAT from midlife data can be used as a surrogate to assess the effect of VAT on incident diseases associated with obesity, as illustrated for postmenopausal breast cancer.

## Introduction

Obesity can present as heterogeneous phenotypes of body fat distribution [1]. Of particular importance is the excess body fat in the form of visceral adipose tissue (VAT) in the intra-abdominal region [2], which is known to induce greater metabolic dysfunction than subcutaneous adipose tissue, thus posing as a greater risk for cardiovascular disease, diabetes, and obesity-related cancers [1]. VAT is not highly correlated with, or well approximated by, body weight, body mass index (BMI), total fat mass, or waist size [3, 4]. Accurate VAT quantification requires 3-dimensional imaging, such as magnetic resonance imaging (MRI) or computed tomography (CT), which is challenging to include in large cohort studies. Consequently, prospective analyses of the role of VAT among pre-symptomatic individuals on subsequent disease risks have been scarce [5–7]. In the Framingham cohorts, a CT-quantified VAT increase by 1 standard deviation (SD) was associated with about 40% higher risk for subsequently developing cardiovascular disease or total incident cancers in multivariate models adjusting for BMI and other risk factors [5]. Similarly, VAT increase per SD was associated with 40% higher BMI-adjusted risk of multiple obesity-related cancers in men of the Health ABC cohort [6]. Recently, VAT assessed at the time of cancer diagnostic imaging has been shown to have a stronger association than BMI with poorer prognosis and survival of breast, colorectal and kidney cancers and cardiovascular disease [8, 9]. Thus, the research to date is strongly supportive for the impact of VAT on metabolic diseases and warrants further studies of early assessment and intervention.

In an MRI study nested within the Multiethnic Cohort (MEC), we previously reported that the correlation of measured VAT with body weight, adulthood weight gain, or total fat mass varied by up to 45% in men and 73% in women when comparing the highest vs. lowest coefficients across five racial/ethnic groups [10]. Also, VAT likely mediated the racial/ethnic disparity in the prevalence of the metabolic syndrome even after adjusting for total fat mass [10]. To fill the gap of limited pre-clinical VAT data, in the same study, we developed a VAT score that provided very good to excellent prediction of VAT based on concurrent measurements of height, BMI, and blood concentrations of nine common blood markers of metabolism and nutrition [11]: $R^2$ for VAT was 0.64 in men and 0.67 in women; area under the receiver operating characteristic curve (AUROC) for visceral obesity (VAT area >150 cm$^2$) was 0.90 in men

and 0.86 in women. We also tested the VAT score in relation to two common obesity-related cancers, of the breast and colorectum, in prospective case-control data nested in the MEC [11]. Although the VAT score was not associated with the risk of colorectal cancer, it was associated with the subsequent risk of postmenopausal breast cancer (odds ratios for higher tertiles = 1.09 (0.86–1.39) and 1.48 (1.16–1.89); p-trend = 0.002), independently of BMI and other breast cancer risk factors. These findings provided proof of concept that our VAT score can serve as a proxy for the concurrent level of VAT.

In the current study, using a subset of the MEC participants for whom data collected 10 years prior to the MRI assessment were available on anthropometry, blood biomarkers, and detailed lifestyle, we examined whether the VAT score based on past data can predict future VAT; we investigated the previously developed VAT score, as well as scores resulting from refitting the prediction model and from enhancing the model by incorporating lifestyle information, such as the quality of habitual diet that was found to be inversely associated with VAT [12]. We also compared the prediction performance of our scores with that of the visceral adiposity index (VAI) [13] and the Metabolic Score for Visceral Fat (METS-VF) [14], two frequently cited and applied visceral fat scores [15, 16].

## Methods

### Study participants

The MEC is an ongoing, population-based cohort following >215,000 men and women of five racial and ethnic groups (African Americans, Japanese Americans, Latinos, Native Hawaiians and Whites) in Hawaii and Los Angeles County, California, who were aged 45–75 years at enrollment in 1993–1996 [17]. A nested case-control dataset was created for breast cancer (n = 950 pairs), identified in area SEER registries, using an incidence density approach by identifying cases with the first diagnosis of postmenopausal invasive breast cancer after the blood collection in MEC (Biorepository, 2001–2006) and selecting a control for each case matched on area, birth year, sex, race/ethnicity, date of blood collection, and hours of fasting [11]. Similar approaches were taken to create a nested case-control dataset for colorectal adenocarcinoma (n = 831 pairs). The Adiposity Phenotype Study (APS) of body composition across the five racial/ethnic groups was conducted on a cross-sectional subset of the MEC in 2013–2016 by re-recruiting a sex-, race/ethnicity- and BMI-stratified sample of 1,861 men and postmenopausal women, aged 60–77 [10]. Exclusions were made for reported BMIs outside the range of 18.5–40 kg/m$^2$, smoking in the past 2 years, body implants or amputation, use of insulin or thyroid medication, and advanced medical conditions. The APS participants, during their visit to the study clinic at the University of Hawaii (UH) or the University of Southern California (USC), underwent a fasting blood collection and anthropometric assessment (height, weight, and circumferences of the waist and hip) and completed study questionnaires. Total fat mass (kg) was determined by a whole-body dual energy X-ray absorptiometry (DXA) scan. An abdominal MRI scan was acquired on 3-Tesla scanners to quantify VAT areas (cm$^2$) at four intervertebral segments of the intra-abdominal cavity (L1–L2, L2–L3, L3–L4, L4–L5) using an axial gradient-echo sequence with breath holds [18]. The average VAT area across the four segments was used as the outcome of interest. The MEC-APS study protocol was approved by the Institutional Review Boards of the UH and USC. Informed consent forms were signed by all participants.

### Circulating biomarker analysis

For the MEC Biorepository (2001–2006), blood samples were collected from ~70,000 MEC participants after an overnight fast; the blood samples were kept refrigerated and processed

within 4 hours into components, which were stored at -186˚C in vapor phase liquid nitrogen [19]. In the APS, fasting blood samples were kept refrigerated and processed within 5 hours into components, which were stored at -80˚C. The APS blood samples were analyzed for 48 biochemical markers (S1 Table) that were selected in a stepwise manner from ~100 established markers of glucose and lipid metabolism, inflammation, nutrition, and hormonal status based on their prediction of visceral and liver fat, as described previously [11]. Blood concentrations of the biomarkers were determined in the UH Cancer Center Analytical Biochemistry Shared Resource laboratory following established protocols [11, 20].

For the current analysis, 500 APS participants (denoted herein as the APS-500) were randomly selected, balanced by sex-race/ethnicity-BMI, and their MEC Biorepository samples (2001–2006) were analyzed for the 48 blood biomarkers for the prediction of future VAT measured in the APS (2013–2016) after 10.6 (mean; SD = 1.2) years from blood collection. The same 48 biomarkers were also analyzed in the MEC Biorepository samples of an independent group of MEC participants selected for the nested case-control analyses of breast cancer and colorectal cancer. The coefficient of variation in blind QCs for the 48 biomarkers ranged between 0.6%-20% [11].

S1 Fig describes the subsets of MEC data used for the current analyses, to predict future VAT and apply the prediction scores to nested case-control studies to examine predicted VAT in relation to risk of incident breast and colorectal cancers.

## Reported anthropometry and diet

We used the data from the MEC 10-year follow-up questionnaire (QX3; 2003–2008), collected around the time of the MEC Biorepository, for this prediction analysis. The MEC baseline questionnaire (QX1) asked the participants to report their weight and height. The 10-year follow-up questionnaire (QX3) asked the participants to report their weight again and to report their waist and hip circumferences measured using an enclosed tape measure. Also, participants were asked to complete a validated, quantitative food frequency questionnaire for usual dietary intake in the past year, from which food/nutrient intakes and diet quality indices were computed [12]. Specifically, the Healthy Eating Index (HEI-2015), a measure of adherence to the 2015 dietary guidelines for Americans, was calculated for the total score (range: 0–100) to indicate overall diet quality and for the 13 component scores to indicate adequacy or moderation of whole fruits, total fruits, whole grains, dairy, total protein foods, seafood and plant proteins, greens and beans, total vegetables, fatty acids, refined grains, sodium, added sugars, and saturated fats [21].

## Statistical analysis

**Original concurrent VAT score.** As described in detail previously [11], the original VAT score was developed for the prediction of concurrent VAT by averaging across 100 regularized regression models that regressed log VAT area on anthropometric and blood biomarker measures in the APS participants separately for men and women. Specifically, elastic net regression was used with the mixing parameter alpha = 0.9 to adjust the contributions of the LASSO ($L_1$) and ridge regression ($L_2$) penalties. This technique shrinks coefficients of unimportant variables to zero; lambda, the constraint parameter, was estimated using 10-fold cross-validation [22]. The variables and parameters included for the original concurrent VAT prediction are provided in S2 Table. For the original prediction score for concurrent VAT, commonly available information on demographics (sex, age, race/ethnicity), simple anthropometry associated with low measurement error (weight, height, BMI), and blood biomarkers were prioritized. For the improvement of the score to predict future VAT, we refit the prediction model using

the above strategy and examined any enhancement by including information on prevalent medical history and important dietary and physical activity practices.

**Forecasted VAT scores.** Forecasted VAT scores were computed for each of the APS-500 subset participants based on their *past* data, i.e., the self-reported anthropometry data from QX3 and the 48 biomarkers analyzed in the MEC Biorepository blood samples. First, we applied the <u>original</u> VAT prediction equation, described in S2 Table, to the data on the anthropometric and nine biomarker predictors. Additionally, we <u>refit</u> sex-specific elastic net regression prediction models of log VAT on the past data, using the same statistical approach and including the same candidate anthropometry and 48 biomarker variables (S1 Table) as in the concurrent VAT score development [11]. The regression coefficients for the variables selected in the refit models were the average of the parameters from models run on 100 bootstrap samples, to avoid overfitting. S3 Table shows the regression coefficients in natural and standardized units. Predictors were retained for the final forecasted VAT models if their standardized coefficient was >0.03 in absolute value in one of the initial sex-specific models, using an *a priori* cutpoint for importance. However, circulating total, HDL and LDL cholesterol were entered together as the importance of each variable in the model was dependent on which component entered first. We also explored whether the forecasted VAT score could be <u>enhanced</u> by including additional information on lifestyles and common metabolic conditions from QX3. Specifically, we considered dietary intake (continuous variables for the amount of macro- and micronutrient consumption and for the total and component HEI-2015 diet quality scores), smoking history (current, former vs. never), and chronic conditions (ever vs. never diagnosed for hypertension or diabetes). The same model selection strategies were applied as described above, resulting in the final sex-specific enhanced models in S4 Table.

**Predicted VAT vs. measured VAT.** We evaluated the performance of the concurrent VAT score and the three forecasted VAT scores (one using the original VAT equation on past data, one using the refit equation on past data, and one using the equation enhanced with past lifestyle and metabolic conditions) in reference to the measured VAT. The $R^2$ statistic was computed as a measure of model fit, i.e., the square of the correlation between each VAT score and measured log VAT area, by sex and sex-racial/ethnic group. Additionally, the AUROC was calculated to determine the agreement between each VAT score and visceral obesity, defined as measured VAT area > 150 cm$^2$ [11]. Sensitivity analyses were performed to evaluate model fit by smoking status (ever vs. never), excluding diabetics at QX3, or excluding women < 60 years of age at QX3 to remove any women who likely experienced recent peri-menopausal changes in adiposity. We also computed the VAI score [13] and the METS-VF score [14], by applying their published equations to the APS-500 participants' past data (QX3/Biorepository) or concurrent data at APS on BMI, waist circumference and blood measures of triglycerides and HDL cholesterol.

**Forecasted VAT scores in association with incident breast and colorectal cancer.** Forecasted VAT scores were also computed for MEC participants selected for the nested case-control analysis, independent of the APS-500 participants, similarly using the anthropometry and lifestyle data from QX3 and the biomarkers analyzed in the MEC Biorepository, as described previously [11]. The VAT score-cancer association was evaluated in a logistic model regressing cancer status on the VAT score, divided into tertiles, with adjustment for BMI and other cancer risk factors for breast or colorectal cancer. Here, the VAT score was adjusted for BMI using the method of residuals, to minimize co-linearity [11]. These analyses were repeated and compared for the forecasted VAT scores using the original, refit and enhanced prediction models.

The R statistical environment (version 3.5.0 (2018-04-23); R Core Team) and the R-package glmnet [22] were used to develop and test the forecasted VAT models (original, refit and

**Table 1. Characteristics of study participants.**

| | Men | | Women | |
|---|---|---|---|---|
| N | 235 | | 244 | |
| **Race/ethnicity, n (%)** | | | | |
| African American | 45 (19%) | | 47 (19%) | |
| Native Hawaiian | 48 (20%) | | 50 (21%) | |
| Japanese American | 46 (20%) | | 48 (20%) | |
| Latino | 46 (20%) | | 49 (20%) | |
| White | 50 (21%) | | 50 (21%) | |
| | QX3 (2003–2008) | APS (2013–2016) | QX3 (2003–2008) | APS (2013–2016) |
| Age, mean years (SD) | 59.2 (2.8) | 69.0 (2.7) | 58.8 (2.9) | 68.5 (2.6) |
| Height, mean meter (SD) | 1.75 (0.08) | 1.73 (0.07) | 1.62 (0.07) | 1.59 (0.07) |
| Body mass index, mean kg/m$^2$ (SD) | 27.5 (4.3) | 28.0 (4.6) | 27.4 (4.8) | 28.0 (5.2) |
| Smoking (ever), % | 48% | 47% | 34% | 31% |
| HEI-2015, mean total score (SD) | 68.9 (10.5) | 70.4 (10.3) | 71.4 (10.6) | 72.4 (10.4) |
| Hypertension, % | 44% | 52% | 39% | 53% |
| Diabetes, % | 10% | 18% | 10% | 17% |
| Total fat mass, mean kg (SD) | | 23.2 (7.6) | | 27.7 (8.8) |
| VAT area, mean cm$^2$ (SD) | | 210 (98) | | 130 (62) |
| VAT area >150cm$^2$, % | | 71% | | 35% |

Abbreviations: APS (The Adiposity Phenotype Study); HEI (Healthy Eating Index); N (number); QX (MEC Questionnaire); SD (standard deviation); VAT (visceral adipose tissue).

The characteristics are presented in mean (standard deviation (SD)) for continuous traits and in percent for categorical traits. Height data were from self-report at cohort baseline for QX3 data analysis and based on technician measurements in the APS study.

enhanced). SAS (version 9.4; SAS Institute, Inc.) was used for additional evaluation of the performance of the prediction scores in relation to measured visceral fat or cancer incidence.

## Results

Table 1 provides the characteristics of the APS-500 participants at the time of MEC 10-year follow-up (QX3; 2003–2008) and APS study (2013–2016). By design, the participants were evenly distributed by sex and race/ethnicity, and men and women were similar in the mean age (59 at QX3 and 69 at APS) and BMI (27–28 kg/m$^2$ at QX3 and APS). Half of the men and one-third of the women were ever smokers. Overall diet quality, as determined by the total HEI-2015 score, improved slightly between QX3 and APS as previously reported [12], whereas the proportion of participants with hypertension or diabetes increased. As reported for the entire APS [11], women had more total fat mass, while men had greater visceral adiposity. Also, visceral adiposity varied significantly by race/ethnicity, highest in Japanese Americans (234 cm$^2$ in men, 176 cm$^2$ in women), lowest in African Americans (161 cm$^2$, 102 cm$^2$), and intermediate for Native Hawaiians, Latinos and Whites (p<0.0001) [11].

Table 2 displays the goodness-of-fit statistics for the prediction of measured VAT using the concurrent and the 3 forecasted (past) VAT scores, as well as using past BMI or BMI and height information only and using VAI and METS-VF. The previously published R$^2$ for VAT prediction using the concurrent VAT score was 0.70 for men (AUROC for visceral obesity = 0.94) and 0.68 for women (AUROC for visceral obesity = 0.86). The R$^2$ for VAT prediction by race/ethnicity ranged from 0.54 for African Americans to 0.78 for Japanese Americans among men, and from 0.57 for African Americans to 0.77 for Native Hawaiians among

**Table 2. Prediction of VAT measured at APS (2013–2016) using VAT prediction scores based on concurrent or past data.**

| | Men | | | | Women | | | |
|---|---|---|---|---|---|---|---|---|
| **VAT prediction** | **Concurrent** | **Forecasted** | | | **Concurrent** | **Forecasted** | | |
| **Data/predictor source** | **APS (2013–2016)** | **QX3 and Biorepository (2001–2008)** | | | **APS (2013–2016)** | **QX3 and Biorepository (2001–2008)** | | |
| **VAT prediction model (equation details)** | **Original** | **Original** | **Refit** | **Enhanced** | **Original** | **Original** | **Refit** | **Enhanced** |
| | (S2 Table) | (S2 Table) | (S3 Table) | (S4 Table) | (S2 Table) | (S2 Table) | (S3 Table) | (S4 Table) |
| VAT score, mean (SD) | 5.21 (0.44) | 5.68 (0.42) | 5.22 (0.40) | 5.21 (0.40) | 4.77 (0.44) | 4.84 (0.43) | 4.74 (0.37) | 4.74 (0.39) |
| $R^2$ all | 0.70 | 0.54 | 0.61 | 0.62 | 0.68 | 0.48 | 0.51 | 0.55 |
| $R^2$ by race/ethnicity | | | | | | | | |
| African Americans | 0.54 | 0.38 | 0.52 | 0.53 | 0.57 | 0.42 | 0.45 | 0.50 |
| Native Hawaiians | 0.76 | 0.67 | 0.70 | 0.64 | 0.77 | 0.52 | 0.60 | 0.58 |
| Japanese Americans | 0.78 | 0.69 | 0.71 | 0.71 | 0.74 | 0.61 | 0.64 | 0.66 |
| Latinos | 0.70 | 0.56 | 0.57 | 0.61 | 0.72 | 0.58 | 0.59 | 0.61 |
| Whites | 0.76 | 0.61 | 0.65 | 0.69 | 0.71 | 0.42 | 0.50 | 0.49 |
| AUROC for visceral obesity (VAT >150 cm$^2$) | 0.94 | 0.88 | 0.89 | 0.90 | 0.86 | 0.82 | 0.82 | 0.85 |
| $R^2$ based on BMI alone | 0.60 | 0.47 | | | 0.51 | 0.40 | | |
| $R^2$ based on BMI, BMI$^2$, height, and height$^2$ | 0.63 | 0.48 | | | 0.56 | 0.42 | | |
| $R^2$ based on VAI | 0.21 | 0.11 | | | 0.29 | 0.17 | | |
| $R^2$ based on METS-VF | 0.72 | 0.44 | | | 0.58 | 0.41 | | |

Abbreviations: APS (Adiposity Phenotype Study); METS-VF (metabolic score for visceral fat); VAI (visceral adiposity index); VAT (visceral adipose tissue).

women. This represented a substantial improvement over prediction based on BMI alone ($R^2$ = 0.60 for men, 0.51 for women), four anthropometry terms alone ($R^2$ = 0.63 for men, 0.56 for women), or VAI ($R^2$ = 0.21 for men, 0.29 for women). The fit for METS-VF was similar to our VAT prediction score for men ($R^2$ = 0.72), but lower for women ($R^2$ = 0.58), similar in magnitude to anthropometry alone.

The VAT scores based on past data resulted in reasonable prediction of future measured VAT (Table 2). The forecasted score using the <u>original</u> prediction equation (S2 Table) had an $R^2$ of 0.54 for men, with a range of 0.38 for African Americans to 0.69 for Japanese Americans, and 0.48 for women, with a range of 0.42 for African Americans and Whites to 0.61 for Japanese Americans. The prediction results were less optimal than the prediction by the concurrent VAT score but represented an improvement compared to prediction using past measures by BMI alone ($R^2$ = 0.47 for men, 0.40 for women), four anthropometry terms ($R^2$ = 0.48 for men, 0.42 for women), VAI ($R^2$ = 0.11 for men, 0.17 for women) or METS-VF ($R^2$ = 0.44 for men, 0.41 for women).

The prediction further improved when the models were <u>refit</u> with the same anthropometry and biomarker variables as in the original model development (Table 2). The optimized refit model retained 16 predictor terms, including the four anthropometry terms (S3 Table). Some of the biomarkers were retained (adiponectin, cholesterol, leptin, total carotenes), some removed (insulin, triglycerides, sex hormone binding globulin), and some added (ALT, glucose, CoQ10, alpha-tocopherol, total anhydro-lutein and total cryptoxanthin) compared to the original prediction model. The refit forecasted VAT scores improved the $R^2$s to 0.61 for men and 0.51 for women, from the previous 0.54 and 0.48, respectively. In particular, $R^2$ for African American men improved from 0.38 to 0.52 by refitting the prediction model.

When the models were <u>enhanced</u> by including past lifestyle and cardiometabolic disease history in addition to the 16 predictors of the refit model above, the optimized enhanced model additionally retained 3 HEI-2015 component scores (dairy, sodium, refined grains) (S4

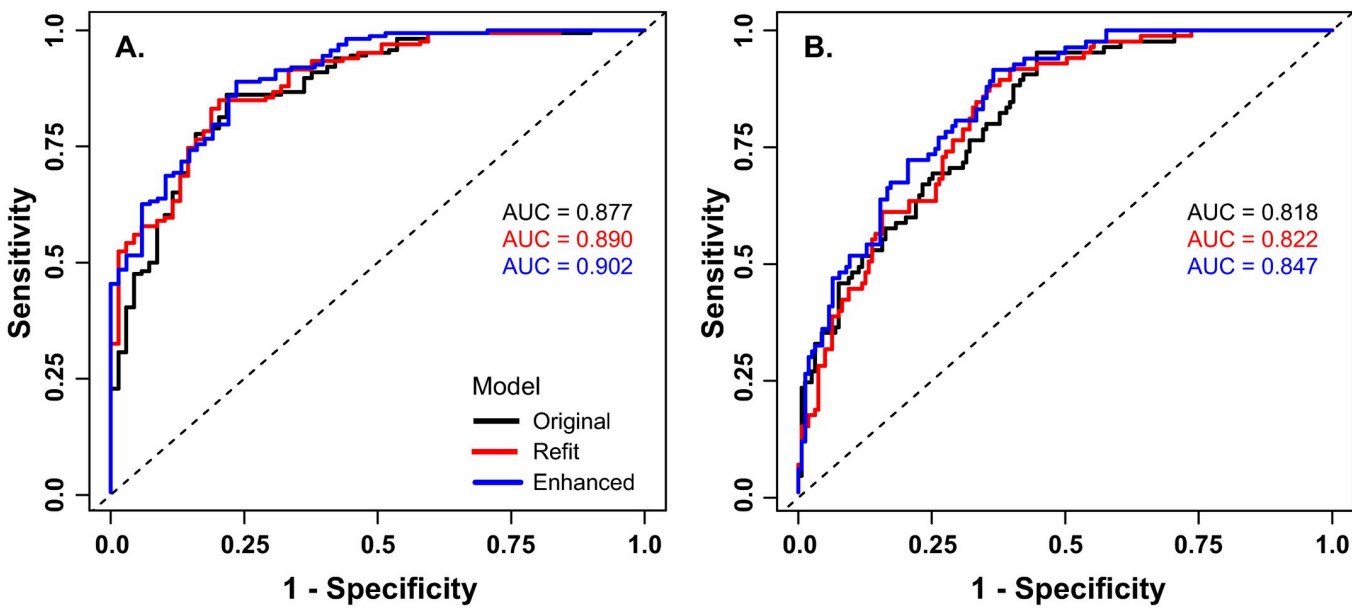

**Fig 1. ROC curves for the classification of visceral obesity for the forecasted VAT prediction models.** (A) For men, ROC curves for the original (black curve), refit (red) and enhanced (blue) models. Text gives the corresponding area under the curve. (B) For women, ROC curves for the original (black curve), refit (red) and enhanced (blue) models. Text gives the corresponding area under the curve.

Table). The enhanced forecasted VAT scores led to improved $R^2$s of 0.62 for men, with a range of 0.53 for African Americans to 0.71 for Japanese Americans, and 0.55 for women, with a range of 0.49 for Whites to 0.66 for Japanese Americans (Table 2).

The AUROC was computed to evaluate agreement between the forecasted VAT prediction scores and future visceral obesity, defined as VAT area > 150 cm$^2$ measured in the APS. Fig 1 shows the ROC curves for three forecasted VAT models separately for men and women. For men, the enhanced model achieved better sensitivity for high specificity (1 −specificity≈0) than the other models, while the models performed similarly for lower specificities (Fig 1(A)). For women, the enhanced model performed slightly better as a classifier for visceral obesity (Fig 1(B)).

We further evaluated the model fit by stratifying on smoking history and by excluding diabetics or younger and likely peri-menopausal women whose adiposity changes might have differed from the others. In men, the prediction accuracy was similar or slightly lower for ever smokers than never smokers (refit model: $R^2$ = 0.60 for ever smokers, 0.61 for never smokers; enhanced model: $R^2$ = 0.61 and 0.63). In contrast, for women, $R^2$s were higher for ever smokers than never smokers (refit model: $R^2$ = 0.58 for ever smokers, 0.48 for never smokers; enhanced model: $R^2$ = 0.58 and 0.53). The model fit slightly improved when participants with diabetes were excluded (refit model: $R^2$ = 0.62 vs. 0.61 for men, 0.53 vs. 0.51 for women; enhanced model: $R^2$ = 0.64 vs. 0.62 and 0.55 vs. 0.55). When the analysis for women was limited to those older than 60 years at 10-year follow-up (39% of all women), the $R^2$ for the refit model was reduced to 0.47 vs. 0.51.

Table 3 presents the association of VAT scores in relation to the subsequent risk of breast cancer in the independent 950 case-control pairs within MEC. We previously presented the results for the original VAT score [11]. Both refit and enhanced VAT scores were positively associated with the subsequent risk of incident breast cancer, after adjusting for risk factors and BMI. The highest tertile, compared to the lowest, was associated with 26–27% higher risks.

**Table 3. Prospective association of VAT prediction scores with incident postmenopausal breast cancer in the Multiethnic Cohort nested case-control analysis (950 case-control pairs).**

| | Breast Cancer | |
| --- | --- | --- |
| | Multivariate-adjusted OR (95% CI) | Multivariate and BMI-adjusted OR (95% CI) |
| **Forecasted VAT score** | | |
| **Original model** | | |
| Tertile 1 (low) | 1.0 (Ref.) | 1.0 (Ref.) |
| Tertile 2 | 1.10 (0.88–1.38) | 1.09 (0.86–1.39) |
| Tertile 3 (high) | 1.45 (1.15–1.82) | 1.48 (1.16–1.89) |
| P-trend | 0.002 | 0.002 |
| **Refit model** | | |
| Tertile 1 (low) | 1.0 (Ref.) | 1.0 (Ref.) |
| Tertile 2 | 1.15 (0.92–1.44) | 1.20 (0.95–1.52) |
| Tertile 3 (high) | 1.26 (1.00–1.58) | 1.27 (1.00–1.62) |
| P-trend | 0.019 | 0.015 |
| **Enhanced model** | | |
| Tertile 1 (low) | 1.0 (Ref.) | 1.0 (Ref.) |
| Tertile 2 | 1.08 (0.86–1.35) | 1.04 (0.82–1.32) |
| Tertile 3 (high) | 1.26 (1.00–1.59) | 1.27 (0.99–1.62) |
| P-trend | 0.025 | 0.012 |
| **VAI** | | |
| Tertile 1 (low) | 1.0 (Ref.) | 1.0 (Ref.) |
| Tertile 2 | 1.13 (0.87–1.47) | 1.16 (0.88–1.54) |
| Tertile 3 (high) | 1.17 (0.90–1.53) | 1.21 (0.92–1.60) |
| P-trend | 0.38 | 0.38 |
| **METS-VF** | | |
| Tertile 1 (low) | 1.0 (Ref.) | 1.0 (Ref.) |
| Tertile 2 | 1.00 (0.77–1.30) | 1.02 (0.78–1.35) |
| Tertile 3 (high) | 1.10 (0.84–1.44) | 1.15 (0.87–1.52) |
| P-trend | 0.51 | 0.50 |

Abbreviations: CI (confidence interval); METS-VF (metabolic score for visceral fat); OR (odds ratio); VAI (visceral adiposity index); VAT (visceral adipose tissue).

Multivariate models for breast cancer were adjusted for matching factors (birth year, sex, race/ethnicity, study area, date of blood collection, hours of fasting), age at blood draw, menopausal hormone therapy, pack-years of smoking, moderate to vigorous activity, family history of breast cancer, age at first live birth, type of menopause, number of children, alcohol (g/day), and energy intake (log transformed kcal/day). BMI was additionally adjusted for using the method of residuals.

In contrast, VAI (OR for highest vs. lowest tertile = 1.21 (0.92–1.60); p-trend = 0.38) and METS-VF (OR = 1.15 (0.87–1.52); p-trend = 0.50) were not associated with breast cancer. Similar to the previously reported null association between the original VAT score and colorectal cancer [11], neither the refit forecast score (OR for tertile 3 vs. 1 = 1.04 (0.80–1.35)) nor the enhanced forecast score (OR = 0.98 (0.75–1.29)) were associated with colorectal cancer risk.

## Discussion

In our analyses nested in a prospective multiethnic cohort, we reasonably forecasted visceral fat amounts over a 10-year span using anthropometry and blood biochemical markers, and

enhanced the prediction with the use of dietary intake information. Specifically, we observed that measured VAT could be predicted by past data applied to our previously developed VAT model ($R^2$ = 0.54 in men, 0.48 in women) or to a refit ($R^2$ = 0.61 in men, 0.51 in women) or enhanced ($R^2$ = 0.62 in men, 0.55 in women) model. Their prediction performance was less optimal than by concurrent data ($R^2$ = 0.70 in men, 0.68 in women) but better than by past BMI alone ($R^2$ = 0.47 in men, 0.40 in women) or by past BMI and height only ($R^2$ = 0.48 in men, 0.42 in women). In a nested case-control analysis of incident breast cancer, we observed a 27% increased risk for the highest vs. lowest tertile of predicted VAT using the refit or enhanced score, independently of BMI and other risk factors. These associations for refit and enhanced forecast VAT scores were weaker than the association of the original VAT score (OR = 1.48) [11], but they were stronger than the association of VAI or METS-VF. Since all three of our VAT scores were computed based on the QX3 data and Biorepository biomarkers for the nested case-control analysis, the stronger association using the original model possibly points to the importance of VAT in mid-life around the time of the QX3/Biorepository, predicted by the original score, than VAT later around the time of the APS that the refit and enhanced models were created to predict.

Our findings are consistent with and further reinforces the previous evidence for the feasibility of VAT prediction using a combination of demographics, anthropometry, and blood biomarker data and of testing the association of predicted VAT with obesity-related incident cancers independently of the effect of BMI. Some VAT prediction scores have been developed based on anthropometry only in reference to accurate imaging-quantified VAT, which we could not apply to our MEC cohort data as their computation requires sagittal diameter [23], thigh circumference [24], or CT parameters [25, 26]. However, our concurrent and forecasted VAT scores outperformed VAI, a VAT prediction score based on BMI, waist circumference, triglycerides, and HDL cholesterol [13], and METS-VF in women, based on BMI, waist to height ratio, age, glucose, triglycerides, and HDL cholesterol [14]. Our VAT score outperformed VAI and were comparable to METS-VF among men and slightly better than METS-VF among women in the entire APS data (n = 1,780) (S5 Table). Previous prospective studies of measured VAT in relation to cancer risk included all incident cancers or all obesity-related cancers due to limited sample sizes [5, 6]. Our finding of the significant positive association between predicted VAT and breast cancer risk further demonstrates the validity and utility of the VAT prediction approach. In our previous analysis of the concurrent VAT score and similarly in our current analysis of forecasted VAT scores, we did not find a BMI-independent association of predicted VAT with incident colorectal cancer [11]. Although the reasons for this lack of an association are not clear, others have similarly reported that CT-quantified VAT at the time of cancer diagnosis was not strongly associated with prognosis for colorectal cancer, while it was a significant factor for prognosis of other obesity-related cancer sites, including breast and bladder cancers [8].

Whereas the concurrent VAT score predicted VAT similarly well in men and women, the forecasted VAT scores performed better for future VAT prediction in men than in women. This may be due to the greater increase in visceral fat in women after menopause. A meta-analysis of longitudinal repeat assessment data before and after menopause suggested that women experience increases in all measures of total and regional fat mass, except for a decrease in percent leg fat, after adjusting for changes attributed to normal aging as seen in men, and that the largest proportional increase was in VAT [27]. Our sensitivity analysis by age indicated higher VAT prediction accuracy for women aged 60 or younger at the time of the predictor data collection. We also found differences in the individual predictors for the concurrent vs. forecasted VAT prediction models, with the latter models relying more on blood leptin and lipid-soluble micronutrients and showing improvement by added diet quality information over smoking or

cardiometabolic disease history. Blood levels of leptin also appeared to override the importance of anthropometric indicators for the forecasted VAT prediction in men. Some of the past studies suggested that baseline leptin predicted the subsequent increase in weight and VAT [28], although this was not supported in other studies [29]. We previously reported a significant inverse association of cross-sectional diet quality and its longitudinal change with VAT [12]. This, combined with the prediction performance of circulating lipid soluble micronutrients, which are considered good biomarkers of overall and long-term diet quality [30], supports the role of nutrition in body fat distribution.

The strengths of this analysis include the availability of MRI-quantified accurate VAT, the use of prospectively collected blood biochemistry markers and lifestyle/disease history as predictors, the balanced inclusion of both sexes and diverse racial/ethnic groups, and the nested case-control analysis of cancers. On the other hand, some limitations might have influenced the results. Our past anthropometry data were based on self-reports and could have resulted in misclassification and lower prediction, although, similar to our previous report [31], we again observed strong correlations between self-reported and measured anthropometry data (r for height = 0.97; r for weight = 0.99; r for BMI = 0.97; r for waist circumference = 0.87; r for hip circumference = 0.90). Our analysis population included some women of typical menopausal ages, but we did not have information on individual timing of menopause to adjust for its effect on VAT prediction [32]. Also, the smaller size of the longitudinal analysis data compared to the cross-sectional data for the original equation (500 vs. 1861) might have limited development of optimal prediction models. Finally, a VAT prediction model composed of fewer and more commonly/easily measured blood biomarkers would facilitate its application to fine-tune obesity and cancer research and to better target community- or primary care-based interventions to obese patients with higher risks. Nevertheless, our results indicate a caveat that some of the simpler prediction scores developed in small samples of homogeneous individuals may not be broadly replicable.

## Conclusion

The challenge of VAT assessment in pre-clinical individuals with latent dysmetabolism remains a major obstacle, while there is an increasing need to advance our understanding of the critical impact of VAT on obesity-associated disease etiology and racial/ethnic health disparities [10]. Our prospective study addresses this challenge and contributes to accessible VAT prediction estimation for research and preventive health.

## Supporting information

**S1 Fig. Diagram showing the data sources for current analysis within the Multiethnic Cohort Study.**
(DOCX)

**S1 Table. List of measured and derived blood biomarkers used for refitting prediction models (previously published in Le Marchand et al. 2020).**
(DOCX)

**S2 Table. The original VAT prediction score equation to predict concurrent VAT in the Adiposity Phenotype Study (published in Le Marchand et al. 2020).**
(DOCX)

**S3 Table. Averaged elastic net regression coefficients for the refit prediction models.**
(DOCX)

**S4 Table. Averaged elastic net regression coefficients for the enhanced prediction models.**
(DOCX)

**S5 Table. Prediction of MRI measurement of VAT using concurrent data in all APS participants (n = 1,780 with no missing data on predictors).** Prediction performance is presented for models including BMI alone, BMI and height, the VAT score developed in MEC-APS, the VAI score, and the METS-VF score.
(DOCX)

**S1 File.**
(DOCX)

## Acknowledgments

We thank the Multiethnic Cohort Study participants who generously donated their time and effort for this body imaging study.

We acknowledge the contribution of the study staff members whose excellent performance made this research possible: the Recruitment and Data Collection Core staff at USC (Adelaida Irimian, Chanthel Figueroa, Brenda Figueroa, Carla Flores, Karla Soriano) and UH (Dr. Terri-lea Burnett, Jane Yakuma, Naomi Hee, Clara Richards, Cheryl Toyofuku, Hui Chang, Janice Nako-Piburn, Reid Sakamoto, Sara Sameshima, Pacer Lee, Emmalyn Pilande, Neelima Nuti, Shirley So, Maya Yamane, Juanita Kaaukai); the Data Management and Analysis Core staff at USC (Zhihan Huang) and UH (Maj Earle, Joel Julian, Anne Tome, Yun Oh Jung, Emil Svrcina); phlebotomy and lab technicians at UH (Wileen Mau, Micha Etrata, Thad Park); the Project Administrative Core staff at UH (Eugene Okiyama), and the UH Analytical Biochemistry Shared Resource lab staff (Dr. Xingnan Li, Laurie Custer, Jen Lai, Tricia DeBaun, Melanie Nakatani, Karly Torii, Alyson Omori). We also thank the Body Imaging Core staff for DXA data processing at UCSF (Dr. Bo Fan) and for MRI imaging and data processing at UH (Dr. Steve Buchthal).

## Author Contributions

**Conceptualization:** Lynne R. Wilkens, Loïc Le Marchand, Unhee Lim.

**Data curation:** Kristine R. Monroe, John A. Shepherd, Adrian A. Franke, Thomas Ernst, Unhee Lim.

**Formal analysis:** Lynne R. Wilkens, Ann M. Castelfranco, Unhee Lim.

**Funding acquisition:** Lynne R. Wilkens, Kristine R. Monroe, Bruce S. Kristal, Iona Cheng, Gertraud Maskarinec, Meredith A. Hullar, Johanna W. Lampe, John A. Shepherd, Adrian A. Franke, Thomas Ernst, Loïc Le Marchand, Unhee Lim.

**Methodology:** Lynne R. Wilkens, Ann M. Castelfranco.

**Project administration:** Loïc Le Marchand.

**Resources:** Loïc Le Marchand.

**Supervision:** Lynne R. Wilkens.

**Writing – original draft:** Lynne R. Wilkens, Ann M. Castelfranco, Unhee Lim.

**Writing – review & editing:** Lynne R. Wilkens, Ann M. Castelfranco, Kristine R. Monroe, Bruce S. Kristal, Iona Cheng, Gertraud Maskarinec, Meredith A. Hullar, Johanna W.

Lampe, John A. Shepherd, Adrian A. Franke, Thomas Ernst, Loïc Le Marchand, Unhee Lim.

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
