## [Decision Letter · Decision Letter 0]

4 Dec 2023

PONE-D-23-25506Prediction of Future Visceral Adiposity and Application to Cancer Research: The Multiethnic Cohort StudyPLOS ONE

Dear Dr. Wilkens,

Thank you for submitting your manuscript to PLOS ONE. After careful consideration, we feel that it has merit but does not fully meet PLOS ONE’s publication criteria as it currently stands. Therefore, we invite you to submit a revised version of the manuscript that addresses the points raised during the review process.

We look forward to receiving your revised manuscript.

Kind regards,

Frank T. Spradley

Academic Editor

PLOS ONE

“This research was funded by grants from the US National Institutes of Health: P01 CA168530 (LRW, KRM, BSK, IC, GM, MAH, JWL, JAS, AAF, TE, LLM, UL), U01 CA164973 (LRW, KRM, IC, LLM), P30 CA071789 (LRW, GM, JAS, AAF, LLM, UL), and P30 CA015704 (MAH, JWL)).”

“Dr. Kristal is a consultant for Metabolon, which played no role in this study.  Dr. Ernst is a consultant for KinetiCor, Inc., which played no role in this study.  The other authors declare no potential conflicts of interest.”

Reviewers' comments:

Reviewer's Responses to Questions

**Comments to the Author**

1. Is the manuscript technically sound, and do the data support the conclusions?

Reviewer #1: Yes

Reviewer #2: Yes

2. Has the statistical analysis been performed appropriately and rigorously? 

Reviewer #1: I Don't Know

Reviewer #2: Yes

3. Have the authors made all data underlying the findings in their manuscript fully available?

Reviewer #1: Yes

Reviewer #2: Yes

4. Is the manuscript presented in an intelligible fashion and written in standard English?

Reviewer #1: Yes

Reviewer #2: Yes

5. Review Comments to the Author

Reviewer #1: Wilkens et al. present an interesting study in which they evaluate the prediction value of a previously developed VAT score by the authors, by enhancing it with additional data. The results they present are interesting and can be of interest in big cancer research and epidemiological studies. However, some sections of the paper are a bit confusing to follow at times, and are lacking some information, and I feel that some changes are necessary.

Abstract

In the introduction of the abstract you talk about the past score with 9 biomarkers but in this study you are enhancing it with others, maybe you can mention it somehow.

In your conclusion you say it could help predict in incident diseases, but really you only looked at breast and CC cancer, I think this sentence should be rewritten, since it could be misleading.

Introduction

- Line 40: please include something such as the “previously developed vat score” to improve understanding

- Line 44: could be worth briefly explaining why you chose these two indexes and methodological differences (either here or in the discussion)

Materials & methods:

- A flow chart of the patients included, data obtained at each time point, etc. would be helpful to understand the process better.

- Line 96: did they report height as well? Please specify better what you mean when you discuss “their anthropometry data” (weight, height, waist circumference…).

- Was the anthropometry after 10 years self-reported or collected by a trained professional? It should be clearly stated.

- A list of the 48 biomarkers in supplementary information will make it easier for the reader to know what has been tested.

- I think this section would benefit from a subsection of (other)variables included to clearly specify what other data (other than diet, anthropometry and blood biomarkers) were employed and how they were collected.

- What statistical program/package was employed? Threshold of significance? This section needs improving.

Results:

- In Table 1, why is education there if it is not discussed later, used to adjust and has been published before? Not sure if it is relevant.

- Table 1 footnotes need improving: HEI, VAT…

- Table 3/results: please include n of participants

-

Discussion:

- Sentence 290-293: I think you should specify that you have previously shown this correlation, how it is written it seems you found this correlation in this study.

- Line 297-299: I think this is an important point and should be discussed further. Where and in which populations could this score be applied realistically? What clinical advantage would it give vs. other scores? Since this is the title of the paper (application to cancer research) I think it deserves a bit more discussion.

- I think the final message should be a bit clearer (in line with previous comment).

Although the paper is well-written, sometimes it is a bit confusing, some minor English/typos revision is needed. Examples (but not exhaustive list):

- Abstract: “9” should be spelled out.

- Full stop after “studies” in line 9.

- Line 76: collected from (instead of on)

- Line 105: As previously described, as described elsewhere, etc.. this occurs throughout the paper, needs to corrected.

- Line 206: pre/peri-menopausal? (now per-mernopausal)

- Line 121, 217, 255: remove the “:”, full stop instead or connector or similar. This is repeated throughout the paper.

- Line 233: (0.62, 0.55) needs R2, men, women….

Reviewer #2: The manuscript presents an intriguing analysis that delves into the predictive capabilities of VAT (visceral adipose tissue) scores, derived from a combination of demographics, anthropometric measures, and blood biomarkers, in relation to cancer risk. The study evaluates three VAT prediction models (original, refit, and enhanced) and their associations with incident breast and colorectal cancer within the Multiethnic Cohort (MEC). The manuscript's strength lies in the inclusion of a diverse cohort that meticulously represents various racial and ethnic groups, enhancing the generalizability of the findings. The balanced representation across sexes and ethnicities and prospective design augments the robustness of the study.

First, the study exhibits several strengths, including the use of MRI-quantified VAT, prospectively collected blood biomarkers, and balanced representation across diverse demographic groups. Moreover, the nested case-control analysis provides valuable insights into cancer associations. One remarkable aspect of this study is the comparison of VAT prediction scores with the conventional measures, BMI, and other published VAT prediction models (VAI and METS-VF). The findings suggest that the VAT scores outperform these conventional metrics in terms of predicting VAT and its associations with cancer risk, particularly breast cancer.

Furthermore, the varying prediction performance between men and women and the influence of factors like menopausal status and nutrition underscore the need for further investigation into interactions and the role of diet quality in body fat distribution. This study provides a foundation for exploring these intricate relationships, and examining potential interactions between VAT, predictive variables, and demographic characteristics could be a valuable next step in future studies.

In addition to exploring interactions, future research could focus on addressing the reasons behind the lack of a BMI-independent association with colorectal cancer and its prognosis. These findings may also prompt the evaluation of other cancer sites and their relationships with VAT scores to provide a comprehensive understanding of obesity-related cancer risks.

The manuscript contributes to our understanding of VAT prediction models and their associations with cancer risk. It opens the door to further investigations into interactions, temporal aspects, and the development of more practical prediction models. The findings underscore the importance of considering VAT as a valuable metric for assessing cancer risk beyond BMI.

Considerations:

1) Elastic Net Regression: The use of LASSO twice, first on past data and then on more recent data, allows for the development of prediction models that take into account the influence of various predictors and changes over time. While LASSO is a widely used regularization technique, the manuscript might benefit from considering the incorporation of Elastic Net regression or at least discussion as a possible next step. Elastic Net combines both L1 (LASSO) and L2 (ridge) regularization, striking a balance between feature selection and model fitting. This technique offers the advantage of simultaneously addressing issues related to variable selection and multicollinearity for a possibly more interpretable model.

2) Variable Selection: It is crucial to elaborate on the rationale behind variable selection for the VAT prediction models. As LASSO automatically selects variables, discussing the clinical significance of the included variables and their impact on VAT prediction would enhance the depth of the manuscript.

3) Graph: The tables, although informative, could benefit from a visual representation, such as, a SHAP plot of the LASSO regression coefficients of variable importance or a single graph overlaying ROC curves by sex and model type.

4) Inconsistent Associations: The manuscript observes a significant positive association between predicted VAT and breast cancer risk but not colorectal cancer. A comprehensive technical exploration of the potential reasons for these inconsistencies, involving a deep dive into the biological mechanisms, pertinent factors, or possible reasons for lack of association is warranted.

6. PLOS authors have the option to publish the peer review history of their article (what does this mean?). If published, this will include your full peer review and any attached files.

Reviewer #1: No

Reviewer #2: **Yes: **Linda M. Polfus, PhD

---

## [Author Response · Author response to Decision Letter 0]

23 Feb 2024

Response to Reviewers

PONE-D-23-25506

Prediction of Future Visceral Adiposity and Application to Cancer Research: The Multiethnic Cohort Study

We thank the Editors and the Reviewers for the helpful suggestions. We have revised our manuscript as summarized below to address the suggestions point by point. 

Response to Journal Requirements:

We have used the above listed templates to revise the manuscript, in two files with and without tracked changes.

Thank you. As stated under Availability of data and materials, we are following the MEC study-wide data sharing policy to make the data available to the research community.

“This research was funded by grants from the US National Institutes of Health: P01 CA168530 (LRW, KRM, BSK, IC, GM, MAH, JWL, JAS, AAF, TE, LLM, UL), U01 CA164973 (LRW, KRM, IC, LLM), P30 CA071789 (LRW, GM, JAS, AAF, LLM, UL), and P30 CA015704 (MAH, JWL)).”

We have included in the resubmission Cover Letter that the funders had no role in the study specifics, as advised above.

“Dr. Kristal is a consultant for Metabolon, which played no role in this study. Dr. Ernst is a consultant for KinetiCor, Inc., which played no role in this study. The other authors declare no potential conflicts of interest.”

We have included in the resubmission Cover Letter that there are no updates in our disclosure since the original submission and that the two co-authors’ reported interests do not alter our adherence to all PLOS ONE policies, as advised above.

We have revised the manuscript as below:

o “As reported for Similar to the previously reported null association between the original VAT score and colorectal cancer (11), neither the refit forecast score (OR for tertile 3 vs. 1 =1.04; 0.80-1.35) nor the enhanced forest score (OR = 0.98 (0.75-1.29)) were associated with colorectal cancer risk (data not shown).” (Lines 298-308): Since we are presenting the results in text as above, we have removed the phrase in the revision.

o “However, our concurrent and forecasted VAT scores outperformed VAI, a VAT prediction score based on BMI, waist circumferences, triglycerides, and HDLS cholesterol (13), and METS-VF in women, based on BMI, waist to height ratio, age, glucose, triglycerides, and HDL cholesterol (14). Our VAT scores outperformed VAI and were comparable to, or in women slightly better than, METS-VF in the entire APS data (n=1,861) (S5 Tabledata not shown).”: (Lines 338-342): We are now providing the specific results in the Supporting Information file.

We have moved the below to the Methods section of our revised manuscript.

o (at the end of the Study Participants section under Methods): Ethics approval and consent to participate: “The MEC-APS study protocol was approved by the Institutional Review Boards of the University of Hawaii and the University of Southern California. Informed consent forms were signed by all participants.” (Lines 108-111)

We have included the captions for the Supporting Information files at the end of our revised manuscript and made in-text citations accordingly.

Response to Reviewers' Comments:

Reviewer #1

Wilkens et al. present an interesting study in which they evaluate the prediction value of a previously developed VAT score by the authors, by enhancing it with additional data. The results they present are interesting and can be of interest in big cancer research and epidemiological studies. However, some sections of the paper are a bit confusing to follow at times, and are lacking some information, and I feel that some changes are necessary.

Thank you for noting the value of our approach in applied epidemiologic studies. 

Abstract

In the introduction of the abstract you talk about the past score with 9 biomarkers but in this study you are enhancing it with others, maybe you can mention it somehow.

We have revised the abstract to indicate the enhancement: “Here we evaluated the VAT score for prediction of future VAT and examined if enhancement with additional biomarkers, lifestyle behavior information, and medical history improves the prediction.” (Lines 18-20)

In your conclusion you say it could help predict in incident diseases, but really you only looked at breast and CC cancer, I think this sentence should be rewritten, since it could be misleading.

We have revised the Abstract Conclusion: “Predicted VAT from midlife data can be used as a surrogate to assess the effect of VAT on incident diseases associated with obesity, as illustrated for postmenopausal breast cancer.” (Lines 35-39)

Introduction

- Line 40: please include something such as the “previously developed vat score” to improve understanding

Thank you. We have revised the text as suggested: “we examined whether the VAT score based on past data can predict future VAT; we investigated the previously developed VAT score, as well as scores resulting from refitting the prediction model and from enhancing the model by incorporating lifestyle information” (Lines 77-79)

- Line 44: could be worth briefly explaining why you chose these two indexes and methodological differences (either here or in the discussion)

We have included the rationale in the same sentence: “We also compared the prediction performance of our scores with that of the visceral adiposity index (VAI) (13) and the Metabolic Score for Visceral Fat (METS-VF) (14), two frequently cited and applied visceral fat scores.” (Lines 80-82)

Methods:

- A flow chart of the patients included, data obtained at each time point, etc. would be helpful to understand the process better.

We have included a supplementary figure to illustrate the data sources (S1 Figure) and described the figure in text, “S1 Fig describes the subsets of MEC data used for the current analyses, to predict future VAT and apply the prediction scores to nested case-control studies to examine predicted VAT in relation to risk of incident breast and colorectal cancers.” (Lines 129-133)

- Line 96: did they report height as well? Please specify better what you mean when you discuss “their anthropometry data” (weight, height, waist circumference…).

- Was the anthropometry after 10 years self-reported or collected by a trained professional? It should be clearly stated.

To address the two questions above, we have revised the text to clarify that the anthropometry data from the MEC questionnaires were either reported (weight and height) or self-measured (waist and hip circumferences): “Both tThe MEC baseline questionnaire (QX1) and 10-year follow-up (QX3) questionnaires asked the participants to report their weight and height. The 10-year follow-up questionnaire (QX3) asked the participants to report their weight again and to report their waist and hip circumferences measured using a tape measure included in the mail. Also, participants were asked to complete a validated, quantitative food frequency questionnaire for usual dietary intake in the past year, from which food/nutrient intakes and diet quality indices were computed (12).” (Lines 136-141)

- A list of the 48 biomarkers in supplementary information will make it easier for the reader to know what has been tested.

We include the list of 48 biomarkers as Supporting Information (S1 Table). 

- I think this section would benefit from a subsection of (other) variables included to clearly specify what other data (other than diet, anthropometry and blood biomarkers) were employed and how they were collected.

As previously described in section starting with Forecasted VAT scores, we only included smoking history and chronic condition history other than diet, anthropometry, and blood biomarkers. (Lines 188-191)

- What statistical program/package was employed? Threshold of significance? This section needs improving.

At the end of the Statistical Analysis section, we now elaborate that, consistent with our approach cited in reference #11, we used the R glmnet package to refit and enhance prediction models and in addition used SAS to test the previously and newly modified prediction scores for performance in relation to measured visceral obesity or cancer incidence. (Line 215-218)

Results:

- In Table 1, why is education there if it is not discussed later, used to adjust and has been published before? Not sure if it is relevant.

We have removed education from Table 1. 

- Table 1 footnotes need improving: HEI, VAT…

We have added Abbreviations to Table 1 footnotes. 

- Table 3/results: please include n of participants

We have inserted in Table 3 title and the associated text that the analysis was for 950 pairs of breast cancer cases and matched controls. (Line 293, Line 310) 

Discussion:

- Sentence 290-293: I think you should specify that you have previously shown this correlation, how it is written it seems you found this correlation in this study.

We have edited the text to present new correlations from APS data and cite the similarity to our previously reported results from another MEC study. (Lines 381-383) 

- Line 297-299: I think this is an important point and should be discussed further. Where and in which populations could this score be applied realistically? What clinical advantage would it give vs. other scores? Since this is the title of the paper (application to cancer research) I think it deserves a bit more discussion.

- I think the final message should be a bit clearer (in line with previous comment).

To address the two comments above, we have revised the last paragraph of Discussion, which we believe has strengthened our Conclusion: “Finally, a VAT prediction model composed of fewer and more commonly/easily measured blood biomarkers would facilitate its application in to fine-tune obesity and cancer research and to better target community- orand primary care-based interventions to obese patients with higher riskssettings in relation to various obesity-associated disease risks. Nevertheless, our results indicate a caveat that some of the simpler prediction scores developed in small samples of homogeneous individuals may not be broadly replicable.” “Conclusion: The challenge of VAT assessment in pre-clinical individuals with latent dysmetabolism remains a major obstacle, while there is an increasing need to advance our understanding of the critical impact of VAT on obesity-associated disease etiology and racial/ethnic health disparities [10]. Our prospective study addresses this challenge and contributes to accessible VAT prediction estimation for research and preventive health.” (Lines 387-398 and 400-404)

Although the paper is well-written, sometimes it is a bit confusing, some minor English/typos revision is needed. Examples (but not exhaustive list):

Thank you. We have corrected the below typos and others throughout the manuscript as shown in tracked changes. 

- Abstract: “9” should be spelled out.

- Full stop after “studies” in line 9

- Line 76: collected from (instead of on)

- Line 105: As previously described, as described elsewhere, etc.. this occurs throughout the paper, needs to corrected.

- Line 206: pre/peri-menopausal? (now per-mernopausal)

- Line 121, 217, 255: remove the “:”, full stop instead or connector or similar. This is repeated throughout the paper.

- Line 233: (0.62, 0.55) needs R2, men, women….

Reviewer #2

The manuscript presents an intriguing analysis that delves into the predictive capabilities of VAT (visceral adipose tissue) scores, derived from a combination of demographics, anthropometric measures, and blood biomarkers, in relation to cancer risk. The study evaluates three VAT prediction models (original, refit, and enhanced) and their associations with incident breast and colorectal cancer within the Multiethnic Cohort (MEC). The manuscript's strength lies in the inclusion of a diverse cohort that meticulously represents various racial and ethnic groups, enhancing the generalizability of the findings. The balanced representation across sexes and ethnicities and prospective design augments the robustness of the study.

First, the study exhibits several strengths, including the use of MRI-quantified VAT, prospectively collected blood biomarkers, and balanced representation across diverse demographic groups. Moreover, the nested case-control analysis provides valuable insights into cancer associations. One remarkable aspect of this study is the comparison of VAT prediction scores with the conventional measures, BMI, and other published VAT prediction models (VAI and METS-VF). The findings suggest that the VAT scores outperform these conventional metrics in terms of predicting VAT and its associations with

---

## [Decision Letter · Decision Letter 1]

21 Jun 2024

Prediction of Future Visceral Adiposity and Application to Cancer Research: The Multiethnic Cohort Study

PONE-D-23-25506R1

Dear Dr. Wilkens,

We’re pleased to inform you that your manuscript has been judged scientifically suitable for publication and will be formally accepted for publication once it meets all outstanding technical requirements.

Kind regards,

Omar Yaxmehen Bello-Chavolla, MD, PhD

Academic Editor

PLOS ONE

Additional Editor Comments (optional):

All queries have been adequately addressed.

Reviewers' comments:

Reviewer's Responses to Questions

**Comments to the Author**

1. If the authors have adequately addressed your comments raised in a previous round of review and you feel that this manuscript is now acceptable for publication, you may indicate that here to bypass the “Comments to the Author” section, enter your conflict of interest statement in the “Confidential to Editor” section, and submit your "Accept" recommendation.

Reviewer #2: All comments have been addressed

Reviewer #3: All comments have been addressed

2. Is the manuscript technically sound, and do the data support the conclusions?

Reviewer #2: Yes

Reviewer #3: Yes

3. Has the statistical analysis been performed appropriately and rigorously? 

Reviewer #2: Yes

Reviewer #3: Yes

4. Have the authors made all data underlying the findings in their manuscript fully available?

Reviewer #2: Yes

Reviewer #3: Yes

5. Is the manuscript presented in an intelligible fashion and written in standard English?

Reviewer #2: Yes

Reviewer #3: Yes

6. Review Comments to the Author

Reviewer #2: Authors addressed comments/concerns regarding revision of Figure 1 and clarification of previous study data elements contributing to this work. Advise to accept manuscript.

Reviewer #3: (No Response)

7. PLOS authors have the option to publish the peer review history of their article (what does this mean?). If published, this will include your full peer review and any attached files.

Reviewer #2: **Yes: **Linda M. Polfus, PhD

Reviewer #3: No

---

## [Editor Report · Acceptance letter]

7 Jul 2024

PONE-D-23-25506R1 

PLOS ONE

Dear Dr. Wilkens, 

I'm pleased to inform you that your manuscript has been deemed suitable for publication in PLOS ONE. Congratulations! Your manuscript is now being handed over to our production team.

Kind regards, 

on behalf of

Dr. Omar Yaxmehen Bello-Chavolla 

Academic Editor

PLOS ONE